# Screening for Liver Fibrosis in the General Population: Where Do We Stand in 2022?

**DOI:** 10.3390/diagnostics13010091

**Published:** 2022-12-28

**Authors:** Clémence M. Canivet, Jérôme Boursier

**Affiliations:** 1Service d’Hépato-Gastroentérologie et Oncologie Digestive, Centre Hospitalier Universitaire d’Angers, 49100 Angers, France; 2Laboratoire HIFIH, UPRES EA3859, SFR 4208, Université d’Angers, 49035 Angers, France

**Keywords:** liver fibrosis, advanced fibrosis, NAFLD, primary care, FIB-4, noninvasive liver fibrosis test

## Abstract

Approximately 30% of the worldwide population has at least one risk factor for liver disease. Identifying advanced liver disease before the occurrence of complications remains a difficult challenge in clinical practice, where diagnosis comes too late for many patients, at the time of liver decompensation or palliative hepatocellular carcinoma, with poor short-term prognosis. Noninvasive, blood- or elastography-based tests of liver fibrosis (NITs) have been developed for the early diagnosis of advanced liver fibrosis. Recent population-based studies evaluating the screening of liver fibrosis with these NITs have provided important information on at-risk groups that should be targeted. New measures based on the sequential use of NITs help to better organize the referral of at-risk patients to the liver specialist. However, energizing these measures will require increased awareness of both chronic liver diseases and the use of NITs among non-specialists.

## 1. Introduction

Chronic liver diseases have become a growing health burden [1,2]. Approximately 25% of the worldwide population is currently living with non-alcoholic fatty liver disease (NAFLD), 1.2% with alcohol use disorders (AUDs), 3.5% with chronic hepatitis B, and 1% with hepatitis C [3,4,5]. Liver fibrosis is the main prognostic factor in chronic liver diseases and longitudinal studies have shown that patient morbidity and mortality were significantly increased in patients with advanced liver fibrosis, i.e., septal fibrosis on liver biopsy [6,7,8]. Patients with chronic liver diseases often have no symptoms, normal physical examinations, and non-specific biological abnormalities. Indeed, only a minority of them will develop advanced liver fibrosis and thus their identification in clinical practice is a challenge for physicians. Consequently, chronic liver diseases are often diagnosed far too belatedly, when the patient’s medium-term prognosis has already become poor. At the time of diagnosis, 75% of cirrhosis cases are already decompensated and 78% of hepatocellular carcinoma cases are at the palliative stage [9,10]. Today, in their quest to identify cases of advanced fibrosis, physicians have at their disposal a range of blood or elastography-based tests able to assess liver fibrosis noninvasively [11]. In this review, we aim to present the current state of screening for advanced liver fibrosis. Thereto, we will attempt to address several questions: Which approach (mass or targeted) should be preferred for the screening of advanced liver fibrosis? Which tests could be relevant for it? What would be the ideal patient pathway? Finally, are non-specialists aware of chronic liver diseases and willing to participate in screening?

## 2. Mass Versus Targeted Screening

Several studies have evaluated screening for advanced liver fibrosis in the general population (Table 1) [12,13,14,15,16,17]. Almost all of them included large samples of unselected patients and evaluated advanced fibrosis with vibration-controlled transient elastography (VCTE) at a threshold of 8.0 kPa, as it provides 90% sensitivity for the diagnosis of advanced liver fibrosis [18]. The prevalence of advanced fibrosis was, overall, similar between the studies at about 5–7%. Moreover, in multivariate analyses, these studies reported the same main factors associated with advanced liver fibrosis: diabetes, other metabolic risk factors (obesity, low high-density lipoprotein cholesterol, high triglycerides, and the presence of metabolic syndrome), excessive alcohol use, and elevated liver enzymes [12,13,14,15,16,19,20].

### 2.1. Type 2 Diabetes Mellitus

Demonstrating the importance of this liver risk factor (Table 1), type 2 diabetes mellitus (T2DM) was consistently associated with an increased risk of advanced liver fibrosis in the general population. In the Rotterdam study, the prevalence of T2DM was 9.8% in patients with VCTE <8 kPa but reached 33.7% in those with VCTE ≥8 kPa [15]. Specific studies in T2DM patients from community-based populations and patients not under insulin therapy showed a 7–10% prevalence of advanced fibrosis as evaluated by VCTE (≥9.6 kPa) [21,22]. In tertiary care centers, the prevalence of advanced liver fibrosis, as evaluated by that same VCTE threshold, increased to 17–21% [23,24,25,26,27]. Some studies have suggested that T2DM severity and duration were associated with a higher risk of advanced fibrosis [25,28]. Kwok et al. found that T2DM patients with VCTE ≥9.6 kPa had a longer duration of diabetes as compared to patients with T2DM and VCTE <9.6 kPa [25]. Another study found that every 1% increase in glycated hemoglobin, measured at the time of liver biopsy, was associated with 15% higher odds of increased fibrosis stage [29]. These findings are helpful for refining subpopulations of interest within the larger T2DM population.

Large studies focusing on hard endpoints have confirmed the strong link between T2DM and liver-related complications. The increased risk of death from cancer in diabetics compared to non-diabetics is the highest for liver cancer, while chronic liver diseases account for the third-highest increase in the risk of death from non-cancer and non-vascular disease [30]. For all these reasons, both the American Diabetes Association (ADA) and the European Associations for the Study of Diabetes (EASD), of the Liver (EASL) and of Obesity (EASO) recommend screening for advanced liver fibrosis in all T2DM patients [31,32].

### 2.2. Factors Related to Metabolic Conditions

In patients with metabolic risk factors, T2DM, obesity and the presence of metabolic syndrome have been shown to be the key features associated with VCTE >7 kPa (Table 1) [33,34,35,36]. An independent association between VCTE ≥8.2 kPa and obesity or diabetes has also been reported in the community-based Framingham Heart Study [34]. In this work, elevated liver stiffness was also independently associated with other cardiovascular risk factors, including metabolic syndrome, hypertension, and low high-density lipoprotein cholesterol. The concurrency of several metabolic risk factors acts synergically on the risk of advanced liver fibrosis. For example, in 3076 patients from the general Spanish population, the prevalence of VCTE >9.2 kPa was only 0.4% in patients without risk factors but it grew to 5.0% in those with ≥1 risk factor (among obesity, T2DM, hyperlipidemia, hypertension, metabolic syndrome or alcohol consumption) [14]. In 890 patients from Sicily, the prevalences of VCTE ≥9.6 kPa in patients without risk factors, with genetic risk factors, with T2DM, and with T2DM + genetic risk factors were 3.7%, 7.7%, 11%, and 36%, respectively [37].

### 2.3. Alcohol Use Disorders

Caballeria et al. showed that the prevalence of VCTE ≥8 kPa was 10.3% in subjects with alcohol use disorders (AUD) (>14 units/week (U/w) for women and >21 U/w for men) compared to 5% in patients without [14]. In a specific study on patients referred from primary care with alcohol consumption >14 U/w, advanced liver fibrosis was independently associated with increasing units of alcohol consumed. In that study, there was a five-fold increase in the risk of developing advanced fibrosis in patients who drank >100 U/w compared to those who drink <35 U/w [38]. Moreover, a longitudinal study in a Swedish population-based cohort with biopsy-proven alcohol-related liver disease found a two-fold increase in the risk of mortality and a very high risk of liver-related death compared to individuals without liver disease [39]. These results should encourage physicians to systematically assess alcohol consumption during patient consultations.

The combination of AUD and metabolic factors has been shown to synergically increase the risk of advanced liver fibrosis [40,41,42]. In a study on primary care patients with AUD and/or T2DM and/or obesity, the prevalence of elevated VCTE was 8.9% in obese patients, 10.8% in T2DM, 36.7% in patients having both obesity and T2DM, and 44% in patients with obesity, T2DM and AUD [36]. Furthermore, a study on NAFLD patients in a tertiary center showed that alcohol consumption >7 U/w was associated with higher overall morbidity, and furthermore that the presence of metabolic syndrome in AUD patients was associated with 27% and 47% increases in overall and liver mortalities, respectively [43].

### 2.4. Elevated Liver Enzymes

As expected, elevated liver enzymes (transaminases) are associated with advanced liver fibrosis (Table 1). In Caballeria’s study, aspartate aminotransferase (AST) and/or alanine aminotransferase (ALT) ≥40 IU/L were associated with a two-fold increased risk of VCTE ≥8 kPa independently of T2DM, obesity and dyslipidemia [14]. Furthermore, a 10-point increase in ALT values has been shown to be associated with a 10% increased risk of elevated liver stiffness measurement [16]. In a study on 190 apparently healthy subjects who underwent a medical health check-up, advanced liver fibrosis was observed in 11.5% of the subjects with ALT ≥19 IU/L but only in 2.5% of those with ALT <19 IU/L [17]. However, it is important to underline that liver enzymes offered only low sensitivity, as 43% of the patients with advanced liver fibrosis had normal AST and ALT values, and only 4.2% of subjects with abnormal AST/ALT had liver stiffness measurement <8 kPa [12].

In summary, screening studies performed in the general population have identified the key risk factors of advanced fibrosis. Specific studies have further confirmed the increased prevalence of advanced fibrosis in populations with those key liver risk factors, giving a rationale for a targeted rather than mass screening approach for advanced liver fibrosis.

## 3. What Do the Guidelines Say about Screening for Advanced Liver Fibrosis?

Since 2016, the EASD, EASL, and EASO recommend liver fibrosis screening in patients with T2DM [31]. In its 2021 guidelines, EASL extended the screening to populations with risk factors for liver diseases (Figure 1) [11]. In 2018, the American Association for the Study of Liver Diseases (AASLD) did not recommend the routine screening for NAFLD in high-risk groups because of uncertainties surrounding diagnostic tests and treatment options, along with a lack of knowledge related to long-term benefits and cost-effectiveness of screening [44]. In its last meeting held in November 2022, the AASLD presented its new practice guidelines and now recommends screening for liver fibrosis in case of clinical suspicion of fatty liver disease (Figure 1) [unpublished]. In their collaborative 2021 guidelines, both the American Gastroenterology Association (AGA) and the American Diabetes Association (ADA) recommend the screening of liver fibrosis in at-risk patients because a timely diagnosis of fibrosis can prevent progression to complications (Figure 1) [45].

## 4. Tools for Screening

A liver biopsy is the gold standard for the assessment of liver fibrosis. However, it is an invasive procedure associated with severe complications in 1–3% of cases and a mortality rate of approximately 1 in 10,000 [46,47]. Because of those aspects, not to mention the high cost of the procedure and the large population to be screened, it becomes clear that noninvasive, repeatable, and ideally cheaper alternatives for the assessment of liver fibrosis are highly desirable—and currently available. These noninvasive alternatives exist mainly in the form of blood-based tests (“functional” methods) or radiology-based techniques using elastography (“physical” methods) [11]. Blood tests include either direct or indirect markers of liver fibrosis, the former reflecting impaired liver function with increasing fibrosis levels, and the latter proteins directly linked to the process of liver fibrogenesis and fibrolysis. Blood tests have the advantages of good reproducibility and potentially extensive availability as they can be prescribed by any physician (Table 2). “Simple” blood tests combine indirect biomarkers of fibrosis, cost nothing (except lab work), and involve only easy calculations via smartphone applications [48]. “Specialized” blood tests combine both direct and indirect biomarkers of fibrosis and do incur usually unreimbursed costs, but they provide better accuracy than simple blood tests [49,50,51,52,53].

There are still controversies about whether non-invasive tests, such as blood-based tests and imaging tests could be complete substitutes for pathology tests by liver biopsy. Even if it remains currently the reference for the evaluation of liver lesions, liver biopsy is impaired by sampling variability and suboptimal inter-observed reproducibility between pathologists, which makes this method not a Gold Standard [54,55]. For this reason, it is not possible to assess the true diagnostic accuracy of NITs in cross-sectional diagnostic studies [56,57]. Nevertheless, diagnostic studies have attracted great interest by demonstrating that NITs are well calibrated on liver fibrosis, the main determinant of the liver-related prognosis in NAFLD. Now, the key challenge is to demonstrate that NITs can accurately stratify the risk of liver-related complications in NAFLD as do the histological stages of liver fibrosis. Such achievement will definitely allow the shift from biopsy to management based on NITs results. Prognostic studies performed in the general population and in patients from tertiary care centers have recently shown the good prognostic accuracy of NITs with results comparable to that of liver biopsy [58,59,60,61,62,63]. In addition, as compared to liver biopsy, NITs can be more easily repeated during follow-up and their evolution allows for a more refined assessment of the prognosis of patients [64].

## 5. Simple Blood Tests

### 5.1. FIB-4

Fibrosis-4 (FIB-4) is a simple blood fibrosis test based on AST, ALT, platelets, and age [65]. In tertiary care, values below its low threshold (1.30) exclude advanced fibrosis with a negative predictive value of >90% and good sensitivity [66,67,68,69]. However, values at or exceeding its high threshold (2.67), with a positive predictive value of around 60–70%, are not sufficiently accurate to confirm advanced fibrosis [69,70,71]. Furthermore, about 30% of patients will have FIB-4 scores in the “grey zone” between the low and high thresholds, where no conclusions can be drawn regarding the diagnosis of advanced liver fibrosis [65,69,70]. Therefore, in these two latter situations (≥high threshold and grey zone scores), a second-line confirmation is mandatory.

### 5.2. NFS

The NAFLD-Fibrosis Score (NFS) is a simple blood test specific to NAFLD. Its calculation requires hyperglycemia, age, body-mass index, AST, ALT, platelet count, and albumin [72]. As with FIB-4, NFS scores ≤1.455 (low threshold) exclude advanced fibrosis with a negative predictive value >90% [69,70] but those ≥0.676 (high threshold), with PPVs at 20–40% in a low prevalence setting, lack sufficient accuracy to rule in advanced fibrosis [70]. NFS too has a grey zone between its low and high thresholds into which 30% of patients fall [72]. Therefore, a second-line confirmatory test for NFS scores ≥0.676 or in the grey zone appears to be the most suitable strategy [11]. Importantly, because the variable “hyperglycemia” weighs heavily in the calculation of the score and when looking to identify advanced fibrosis in the T2DM setting, there is a very low rate of patients under the low threshold: 30% in primary care and 3–13% in diabetes clinics [73,74,75]. Consequently, NFS should not be used for the screening of advanced fibrosis in T2DM patients.

## 6. Specialized Test

### 6.1. Elastography-Based Techniques

VCTE with Fibroscan (Echosens, Paris, France) was the first of the various elastography techniques available today and continues to have the most available data [66,76,77]. VCTE quantifies the speed of a mechanically induced shear wave in liver tissue and calculates liver stiffness from it [78]. Several studies have shown that liver stiffness correlates well with the degree of liver fibrosis in chronic liver diseases and especially in NAFLD [52,76]. In direct comparisons, the accuracy of VCTE has been shown to be higher than those of simple and specialized blood tests [52]. In recent studies from over 5,000 biopsy-proven NAFLD patients, VCTE <7–8 kPa had very good, 85%, sensitivity, and VCTE >12 kPa provided 90% specificity for diagnosing advanced fibrosis [18,70]. These results suggest that VCTE is a viable second-line option after simple tests [11,31].

Other elastography-based techniques coupled with ultrasound, such as point shear wave elastography and two-dimensional shear wave elastography, are now available (Table 2). Because these devices are readily available in radiology practices, they represent an interesting option to increase the availability of liver elastography in the context of screening. These techniques are as accurate as VCTE for the diagnosis of advanced fibrosis [76,79,80,81,82,83,84]. However, the consensus remains to be reached on thresholds for disease risk stratification in relation to histology, and reliability criteria await validation [85,86]. Finally, magnetic resonance elastography, based on magnetic resonance imaging, is the most accurate technique for staging liver fibrosis but its use is restricted to clinical research because of its limited availability and high cost [76,87].

### 6.2. Specialized Blood Test

The three most-validated specialized blood tests are the Enhanced Liver Fibrosis (ELF) test, FibroMeter, and Fibrotest. ELF is a panel of three direct markers of fibrosis: type III procollagen peptide, hyaluronate, and tissue inhibitor of metalloproteinase-1 [88,89]. FibroMeter is based on age, sex, AST, platelets, prothrombin time, urea, hyaluronate, and α2-macroglobulin [90], and Fibrotest on age, sex, glutamyl gamma transferase, alpha-2 macroglobulin, haptoglobin, apolipoprotein A1 and bilirubin [54]. Diagnostic studies using liver biopsy as a reference have demonstrated good rule-out sensitivity (80–90%) and good rule-in specificity (90–95%) of these NITs for the diagnosis of advanced liver fibrosis in chronic liver diseases [49,51,71,91,92]. Because these specialized blood tests include more expensive blood markers, they are best reserved for second-line evaluations of liver fibrosis, as recently proposed [93].

## 7. Referral Pathways with Algorithms

NITs have complementary advantages (Table 2). They offer the potential to identify, among the many patients with liver risk factors, those who have developed advanced liver disease and therefore require specialized management. As combined NITs give better results than single ones [94], the patient pathway must be built upon an initial simple test followed by a specialized test to optimize the identification of patients with advanced liver fibrosis (Figure 2) [11,45,93,95,96,97,98,99].

Patients with risk factors for chronic liver diseases (type 2 diabetes mellitus, obesity, metabolic syndrome, elevated liver enzymes, and alcohol consumption) seen in primary care should undergo testing by a simple blood test (FIB-4 or NFS), followed, if positive, by transient elastography or a specialized blood test (ELF, FibroMeter or Fibrotest) before being referred to a liver specialist.

ELF: enhanced liver fibrosis; FIB-4: Fibrosis-4; NFS: NAFLD fibrosis score

## 8. Algorithms in Primary Care

Srivastava et al. evaluated a two-step pathway based on FIB-4 and ELF in new NAFLD patients from primary care. Patients with FIB-4 <1.30 were deemed to be at low risk of advanced fibrosis and remained in primary care. Patients with FIB-4 >3.25 were considered to be at high risk of advanced fibrosis and recommended for referral to secondary care. Patients with FIB-4 in the grey zone had an ELF test (used with the 9.8 threshold) to determine the risk of advanced fibrosis. It should be noted that very few studies have used the 3.25 rule-in threshold (coming from HIV-HCV co-infection [65]) and that international guidelines uniformly propose to use FIB-4 with its 1.30 rule-out and 2.67 rule-in thresholds adapted for NAFLD [67]. Compared to standard care, the pathway proposed by Srivastava et al. resulted in an 81% reduction in unnecessary referrals and improved the identification of patients with advanced fibrosis and cirrhosis [93]. In a Canadian study, Davyduke et al. evaluated an algorithm combining FIB-4 with a threshold of 1.30 and VCTE with a threshold of 8.0 kPa. Of the 597 patients included from primary care with steatosis on ultrasound or elevated ALT, only 4% had FIB-4 ≥1.30 and VCTE ≥8 kPa, demonstrating a significant reduction in the number of patients requiring referral to a liver specialist [98]. Mansour et al. evaluated the FIB-4/VCTE algorithm in 467 T2DM patients from primary care. Among the 85/467 patients with raised FIB-4, 38/58 had a VCTE ≥8 kPa, and 20 were found to have advanced fibrosis (4.5%). Alcohol use (particularly drinking >14/21 U/w) and body-mass index were predictors of advanced liver fibrosis in that study [99]. Cost-effectiveness studies have confirmed that these pathways do reduce costs compared to a refer-all strategy [100,101]. Nevertheless, although such pathways have been widely studied in at-risk patients [11,45,95,96,97], their sequential use in primary care has received little attention [93,98,99]. Additionally, cost-effectiveness analyses have been performed only in the United Kingdom and Italy; similar studies should be encouraged in other countries to validate these results [100,101].

## 9. What Do the Guidelines Say about Referral Pathways Based on Noninvasive Tests?

The EASL and the AGA recommend evaluation of liver fibrosis in patients with AUD or metabolic cofactors (Figure 1) [11,45]. In its last meeting held in November 2022, the AASLD also presented its practice guidelines with an algorithm for liver fibrosis assessment in patients with suspected fatty liver disease [not published] (Figure 1). All three algorithms from EASL, AGA, and AASLD propose as a first step the identification of liver risk factors (alcohol and metabolic disorders). Further steps rely on NITs with first the simple blood test FIB-4 eventually followed by liver stiffness measurement with VCTE and/or a specialized blood test. The thresholds for the different tests used are similar between the three algorithms (Figure 1).

## 10. Awareness of Chronic Liver Diseases

Due to the large population of patients at risk of chronic liver diseases, general practitioners (GPs) and other non-liver specialists need to play a major role in the early diagnosis of advanced liver disease.

However, several studies have reported that GPs have little awareness of chronic liver diseases [102,103,104,105,106]. In a survey sent to 64 GPs, Van Asten et al. showed that the acronyms NAFLD and NASH were not known by, respectively, 34% and 53% of the respondents and that 96% of them never or rarely screened for the corresponding pathologies [107]. Other studies have shown that GPs were often unaware of the existence of NITs (25%) [105] and unsure whether VCTE (36.3%), NFS/FIB-4 (43.1%), or ELF (56.9%) could help monitor disease progression [102]. In a Dutch survey, NITs were never (73%) or rarely (22%) used by GPs to evaluate disease severity [102,107]. Therefore, most patients diagnosed with a risk factor for chronic liver disease are referred to a liver specialist because of non-specific liver enzyme abnormalities rather than abnormal NITs suggesting advanced liver fibrosis [102,107]. Interestingly, however, after a brief explanation of the simple NITs, almost all physicians expressed a willingness to use them in practice [105].

Low awareness is not limited to GPs. A retrospective analysis of the French hospitalizations database found a very low 0.4% prevalence of the NAFLD/NASH diagnosis code among 50 million adult patients [108]. A questionnaire sent to non-liver specialists in two tertiary hospitals in Brisbane (Australia) showed that the respondents were aware of the association of NAFLD with cardiovascular risk factors (90%) and that of NASH with increased overall mortality, but 71% of them did not refer patients to hepatology if NAFLD was suspected [109]. In a survey sent to 133 British diabetologists, only 5.7% had used or would use a noninvasive algorithm to assess the severity of NAFLD [110]. Another survey of 178 French diabetologists showed that 59% of them underestimated the prevalence of chronic liver diseases. Although 97% of the diabetologists were familiar with NITS, only 29% of them cited FIB-4 [105].

Patients themselves have little awareness of chronic liver diseases. In a study performed in the United States that included more than 10,000 adults from the National Health and Nutrition Examination Survey, 96%, 66%, and 44% of adults with, respectively, NAFLD, hepatitis B or C were unaware they had liver disease [111,112]. A survey among the Turkish community in the Netherlands, wherein hepatitis B is an important health problem, revealed nonetheless a low level of awareness of that pathology [113]. Interestingly, an international cross-sectional survey of 1,411 NAFLD patients recently showed that patients who knew their fibrosis stage were more compliant with lifestyle management [114].

Finally, a survey of European experts to gather information on policies, clinical guidelines, awareness, and monitoring, coupled with data extracted from official documents, revealed a general lack of national policies and awareness campaigns on NAFLD [115]. All these results demonstrate the urgent need for hepatology associations to actively raise awareness of chronic liver diseases and alert authorities as to the weight these pathologies have on current and future public health.

## 11. Conclusions

Despite the extensive prevalence of chronic liver diseases in the general population, only a small proportion of patients are diagnosed with advanced fibrosis. Patient pathways based on the sequential use of NITs have been proposed in recent years to optimize healthcare systems and improve the identification of patients with advanced fibrosis. These new approaches require the active participation of GPs and other non-liver specialists and the involvement of public authorities to further energize their use.

## Figures and Tables

**Figure 1 diagnostics-13-00091-f001:**
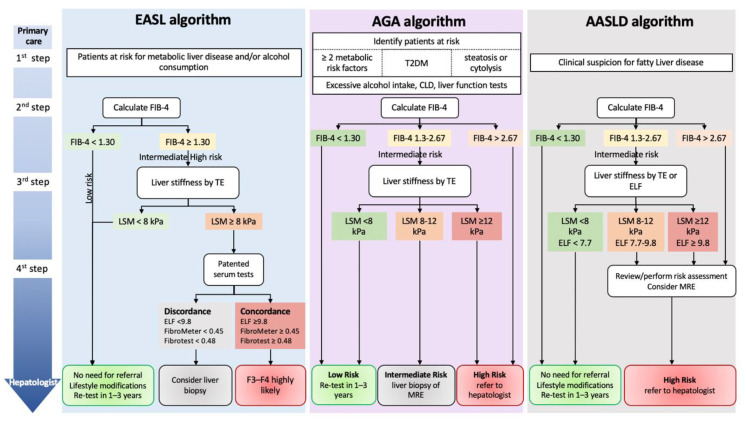
Referral pathway proposed by the European Association for the Study of the Liver (EASL), the American Gastroenterology Association (AGA), and the American Association for the Study of Liver Disease (AASLD) to noninvasively assess advanced liver fibrosis. Modified from [11,45]. CLD: chronic liver disease; ELF: enhanced liver fibrosis; LSM: liver stiffness measurement; MRE: magnetic resonance elastography; T2DM: type 2 diabetes mellitus, TE: transient elastography.

**Figure 2 diagnostics-13-00091-f002:**
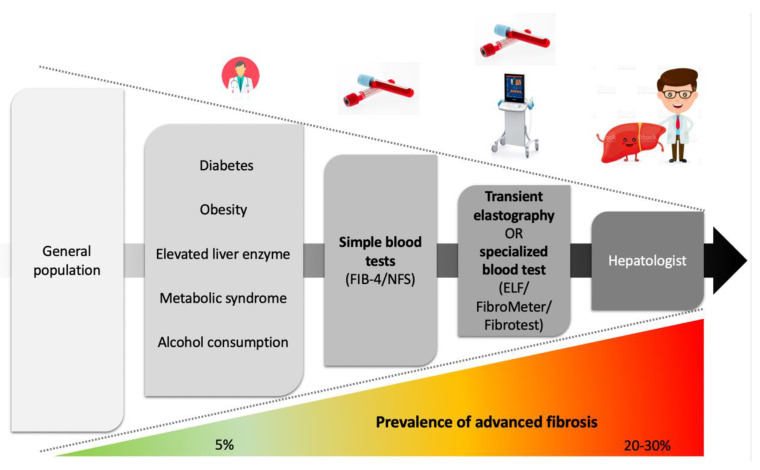
Proposed patient pathway in the general population.

**Table 1 diagnostics-13-00091-t001:** Results of the studies reporting liver fibrosis prevalence in unselected participants with the use of transient elastography in a community setting.

	Study Location	Patients (Years)	Sample Size with TE Available (n)	VCTE Thresholds	Prevalence of Outcome (%)	Risk Factors Independently Associated with Liver Fibrosis in Multivariate Analysis
Roulot 2011 [12]	France	≥45	1190	8.0 kPa	7.5	Metabolic syndrome; BMI ≥30; Age ≥57 years; Diabetes; GGT ≥45 IU/l; ALT ≥40 IU/l
Wong 2012 ^a^ [19]	Hong Kong	18–70	759	9.6 kPa	2.0	ALT level; BMI
Fabrellas 2013 [13]	Spain	18–70	495	6.8 kPa	5.7	-
You 2015 [17]	Korea	adults	159	7.0 kPa	6.9	BMI >24.2; ALT >19 IU/l; carotid intimal media thickness >0.68 mm; ≥1 calcified carotid plaque
Koehler 2016 [15]	The Netherlands	≥45	3041	8.0 kPa	5.6	Type 2 diabetes; liver steatosis; HBsAg and/or anti-HCV positive; age; spleen size; current or former smoking; ALT level
Caballeria 2018 [14]	Spain	18–75	3076	8.0 kPa	5.8	Male sex; AST and/or ALT >ULN; abdominal obesity; glucose level ≥100 mg/dL; low HDL; triglyceride level ≥150 mg/dL; type 2 diabetes
Llop 2021 [16]	Spain	20–79	11,440	8.0 kPa	5.6	Age; male sex; AST level; ALT level; metabolic syndrome
Nah 2021 [20]	Korea	19–85	8183	2.90 kPa *	9.5	Age; male sex; diabetes; HBsAg positivity; abnormal LFT; obesity; metabolic syndrome

ALT: alanine aminotransferase; AST: aspartate aminotransferase; BMI: body-mass index in kg/m^2^; GGT: glutamyl gamma transferase; HBsAg: HBs antigen; HCV: hepatitis C virus; HDL: high-density lipoprotein; LDL: low-density lipoprotein; VCTE: vibration controlled transient elastography. ^a^ Exclusion of chronic viral hepatitis. * by MRE.

**Table 2 diagnostics-13-00091-t002:** Advantages and disadvantages of the main noninvasive liver fibrosis tests from [11].

	Advantages	Disadvantages
Simple blood tests	- Good reproducibility and high applicability - No direct cost and widely available - Well validated - Can be performed in the outpatient clinic	- Not-liver-specific - Performance below that of VCTE and patented serum markers - False positives with FIB-4 and NFS in patients aged >65 years
Specialized blood tests	- Good reproducibility and high applicability - Well validated - Can be performed in the outpatient clinic	- Costly - Not liver-specific - False positives in patients with extrahepatic inflammatory conditions, profibrotic, extrahepatic disease, or others (e.g., hemolysis, Gilbert syndrome)
Transient elastography	- Most widely-used and validated technique - Point-of-care - Well-defined quality criteria - Good reproducibility	- Requires a dedicated device - False positives in patients with acute hepatitis, extrahepatic cholestasis, liver congestion, food intake, and excessive alcohol intake
pSWE/2D-SWE	- Can be combined with regular ultrasound - Performance equivalent to that of VCTE	- False positives in patients with acute hepatitis, extrahepatic cholestasis, liver congestion, food intake, and excessive alcohol intake
MRE	- Can be implemented on a regular MRI machine - Examination of the whole liver	- Not applicable in case of iron overload - Requires an MRI facility - Time-consuming - Costly

VCTE: vibration controlled transient elastography; pSWE: point shear wave elastography; 2D-SWE: bidimensional shear wave elastography; MRE: magnetic resonance elastography; MRI: magnetic resonance imaging; FIB-4: Fibrosis-4; NFS: NAFLD fibrosis score.

## Data Availability

Not applicable.

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
