# Peer review of "Screening for Liver Fibrosis in the General Population: Where Do We Stand in 2022?"

_diagnostics, 2022, doi:10.3390/diagnostics13010091_

Round 1

Reviewer 1 Report

In this literature review entitled `Screening for liver fibrosis in the general population: Where do we stand in 2022?` the authors provide an update regarding current screening strategies for advanced stages of chronic liver disease. The article is well written and easy to follow, giving a good overview. However, I would have preferred to discuss the current American and European guidelines as well as national guidelines more extensively (e.g. chapter 3) and potentially compare screening algorythms. On top, one might expand chapter 2.2 beyond obesity and T2DM as currently cardiovascular risk factors such as arterial hypertension are discussed intensively in the field:

Long MT, Zhang X, Xu H et al. Hepatic Fibrosis Associates With Multiple Cardiometabolic Disease Risk Factors: The Framingham Heart Study. Hepatology 2021; 73: 548-559. doi:10.1002/hep.31608.

Author Response

In this literature review entitled `Screening for liver fibrosis in the general population: Where do we stand in 2022, the authors provide an update regarding current screening strategies for advanced stages of chronic liver disease. The article is well written and easy to follow, giving a good overview.

  1. However, I would have preferred to discuss the current American and European guidelines as well as national guidelines more extensively (e.g. chapter 3) and potentially compare screening algorithms.

We thank the reviewer for his relevant comment. We have updated the Figure 1 (attached) where we now propose a parallel view of the algorithms proposed by 3 international guidelines: EASL, AGA, and the one from AALSD unveiled at the Liver Meeting last November.

We have also updated the manuscript with the following paragraphs:

- in chapter 3: “Since 2016, the EASD, EASL and EASO recommend liver fibrosis screening in patients with T2DM [32]. In its 2021 guidelines, EASL extended the screening to populations with risk factor for liver diseases (Figure 1) [11]. In 2018, the American Association for the Study of Liver Diseases (AASLD) did not recommend the routine screening for NAFLD in high-risk groups because of uncertainties surrounding diagnostic tests and treatment options, along with lack of knowledge related to long-term benefits and cost-effectiveness of screening [45]. In its last meeting that hold in November 2022, the AASLD has presented its new practice guidelines and now recommend the screening for liver fibrosis in case of clinical suspicion of fatty liver disease (Figure 1) [unpublished]. In their collaborative 2021 guidelines, both the American Gastroenterology Association (AGA) and the American Diabetes Association (ADA) recommend the screening of liver fibrosis in at-risk patients because timely diagnosis of fibrosis can prevent progression to complications (Figure 1) [46].”

- in chapter 9: “The EASL and the AGA recommend evaluation of liver fibrosis in patients with AUD or metabolic cofactors (Figure 1) [11,46]. In its last meeting that hold in November 2022, the AASLD has also presented its practice guidelines with an algorithm for liver fibrosis assessment in patients with suspected fatty liver disease [not published] (Figure 1). All three algorithms from EASL, AGA and AASLD propose as a first step the identification of liver risk factors (alcohol, metabolic disorders). Further steps rely on NITs with first the simple blood test FIB-4 eventually followed by liver stiffness measurement with VCTE and/or a specialized blood test. The thresholds for the different tests used are similar between the three algorithms (Figure 1).”

  1. On top, one might expand chapter 2.2 beyond obesity and T2DM as currently cardiovascular risk factors such as arterial hypertension are discussed intensively in the field:

Long MT, Zhang X, Xu H et al. Hepatic Fibrosis Associates With Multiple Cardiometabolic Disease Risk Factors: The Framingham Heart Study. Hepatology 2021; 73: 548-559. doi:10.1002/hep.31608.

We read with interest the article related by the reviewer. We have added a paragraph in the chapter 2.2: “An independent association between VCTE ≥8.2 kPa and obesity or diabetes has also been reported in the community-based Framingham Heart Study [35]. In this work, elevated liver stiffness was also independently associated with other cardiovascular risk factors, including metabolic syndrome, hypertension, and low high-density lipoprotein cholesterol.”

Reviewer 2 Report

In this manuscript, Canivet et al. described that new sequences based on the sequential use of NITs help to better organize the referral of at-risk patients to the liver specialist. However, energizing these measures will require increased awareness of both chronic liver disease and the use of NITs among non-specialists. This reviewer has the following concerns. 

Major comments: 

1. There are still controversies about whether non-invasive tests, such as NITs and imaging tests could be complete substitutes for pathology tests by liver biopsy. The author should discuss the consistency of diagnosis among NITs, imaging tests, and pathology tests, especially in healthy individuals and patients with chronic liver disease. 

2. In 2.1. Type 2 diabetes mellitus, the author described that “In several studies, T2DM severity and duration appeared to be associated with a higher risk of ~”. This information should be concretely mentioned using HbAic and duration.  

3. A high threshold (2.67) of FIB-4 and the “grey zone” must be handled carefully. This threshold is 2.67 in 5.1. FIB-4, but 3.25 in 8. Algorithms in primary care. The reason why this value is different between them should be explained. Also, the author had better describe that FIB-4 is useful for screening in general population, but has a limitation in patients with chronic liver disease. 

Minor comments: 

1.      Page 1, line 3 in INTRODUCTION, “1,2%” should be “1.2%” or “1-2%”.

2.      Page 3, line 6 in 2.4. Elevated liver enzymes, “LSM” should be “liver stiffness measurement (LSM)”

3.      Page 9, “10. CONCLUSION” should be “11. CONCLUSION”.

Author Response

In this manuscript, Canivet et al. described that new sequences based on the sequential use of NITs help to better organize the referral of at-risk patients to the liver specialist. However, energizing these measures will require increased awareness of both chronic liver disease and the use of NITs among non-specialists. This reviewer has the following concerns. 

Major comments: 

  1. There are still controversies about whether non-invasive tests, such as NITs and imaging tests could be complete substitutes for pathology tests by liver biopsy. The author should discuss the consistency of diagnosis among NITs, imaging tests, and pathology tests, especially in healthy individuals and patients with chronic liver disease. 

We thank the reviewer for this important comment. Following the Reviewer comment, we have added the following paragraph in the chapter 4 of the manuscript: “there are still controversies about whether non-invasive tests, such as blood-based tests and imaging tests could be complete substitutes for pathology tests by liver biopsy. Even if it remains currently the reference for the evaluation of liver lesions, liver biopsy is impaired by sampling variability and suboptimal inter-observed reproducibility between pathologists, which makes this method not a Gold Standard [55-56]. For this reason, it is not possible to assess the true diagnostic accuracy of NITs in cross-sectional diagnostic studies [57,58]. Nevertheless, diagnostic studies have had the great interest of demonstrating that NITs are well calibrated on liver fibrosis, the main determinant of the liver-related prognosis in NAFLD. Now, the key challenge is to demonstrate that NITs can accurately stratify the risk of liver related-complications in NAFLD as do the histological stages of liver fibrosis. Such achievement will definitely allow the shift from biopsy to a management based on NITs results. Prognostic studies performed in the general population and in patients from tertiary care centers have recently shown the good prognostic accuracy of NITs, with results comparable to that of liver biopsy [59-64]. In addition, as compared to liver biopsy, NITs can be more easily repeated during follow-up and their evolution allows for a more refined assessment of the prognosis of patients [65].”

  1. In 2.1. Type 2 diabetes mellitus, the author described that “In several studies, T2DM severity and duration appeared to be associated with a higher risk of ~”. This information should be concretely mentioned using HbA1c and duration.  

To better described the link between the characteristics of diabetes and the severity of liver fibrosis, we added the following sentences in the paragraph 2.1: “Some studies have suggested that T2DM severity and duration were associated with a higher risk of advanced fibrosis [28,29]. Kwok et al. found that T2DM patients with VCTE ≥9.6 kPa had a longer duration of diabetes as compared to patients with T2DM and VCTE < 9.6 kPa [28]. Another study found that every 1% increase in glycated hemoglobin, measured at the time of liver biopsy, was associated with 15% higher odds of increased in fibrosis stage [30].”

  1. A high threshold (2.67) of FIB-4 and the “grey zone” must be handled carefully. This threshold is 2.67 in 5.1. FIB-4, but 3.25 in 8. Algorithms in primary care. The reason why this value is different between them should be explained. Also, the author had better describe that FIB-4 is useful for screening in general population, but has a limitation in patients with chronic liver disease. 

We fully agree with the reviewer on these points.

FIB-4 has been developed in patients with HIV-HCV co-infection and the calculated rule-out and rule-in thresholds were 1.45 and 3.25 (Sterling, Hepatology 2006). These thresholds have been reassessed in NAFLD at 1.30 and 2.67 (Shah, Clin Gastroenterol Hepatol 2009), and these latter are now uniformly used by international guidelines (see new figure 1). Very few studies have used the 3.25 rule-in threshold, as did Srivastava et al. To avoid confusion for the reader, we have removed figure 3 and added the following comment in chapter 8 when describing the patient pathway of the Srivastava et al. study. “It should be noted that very few studies have used the 3.25 rule-in threshold (coming from HIV-HCV co-infection [66]), and that international guidelines uniformly propose to use FIB-4 with its 1.30 rule-out and 2.67 rule-in thresholds adapted for NAFLD [68]. Compared to standard care, the pathway proposed by Srivastava et al. resulted in an 81% reduction…”

We agree with the Reviewer that FIB-4 is useful for screening in the general population as it provides excellent negative predictive value in low prevalence settings. We also acknowledge that FIB-4 has insufficient positive predictive value in patients with chronic liver disease, with the need for further specialized evaluation as now emphasized in the new Figure 1 (attached).

Minor comments: 

  1. Page 1, line 3 in INTRODUCTION, “1,2%” should be “1.2%” or “1-2%”.

Thank you, this error is corrected and “1,2%” is changed to “1.2%”

  1. Page 3, line 6 in 2.4. Elevated liver enzymes, “LSM” should be “liver stiffness measurement (LSM)”

We have made the change. As “liver stiffness measurement” is only used twice, we did not use an abbreviation

  1. Page 9, “10. CONCLUSION” should be “11. CONCLUSION”.

This error is corrected
